# Salt-Tolerant Phenomena, Sequencing and Characterization of a *Glyoxalase*
*I* (*Jojo-Gly I*) Gene from Jojoba in Comparison with Other *Glyoxalase I* Genes

**DOI:** 10.3390/plants9101285

**Published:** 2020-09-29

**Authors:** Heba Allah A. Mohasseb, Mohei El-Din Solliman, Ibrahim S. Al-Mssallem, Mohammed M. Ba Abdullah, Ahmed Saud Alsaqufi, Wael F. Shehata, Hany A. El-Shemy

**Affiliations:** 1Plant Biotechnology Department, College of Agricultural and Food Sciences, King Faisal University, P.O. Box 400, Al-Ahsa 31982, Saudi Arabia; hmohasseb@kfu.edu.sa (H.A.A.M.); imssallem@kfu.edu.sa (I.S.A.-M.); wshehata@kfu.edu.sa (W.F.S.); 2Plant Biotechnology Department, National Research Centre, Dokki-Egypt, Giza P.O. Box 12622, Cairo, Egypt; 3Biological Sciences Department, College of Sciences, King Faisal University, P.O. Box 400, Al-Ahsa 31982, Saudi Arabia; mmb303@gmail.com; 4Animal Production Department, College of Agricultural and Food Sciences, King Faisal University, P.O. Box 400, Al-Ahsa 31982, Saudi Arabia; aalsaqufi@kfu.edu.sa; 5Plant Production Department, College of Environmental Agricultural Science, El–Arish University, El–Arish P.O. Box 45511, North Sinai, Egypt; 6Biochemistry Department, Faculty of Agriculture, Cairo University, Giza P.O. Box 12613, Cairo, Egypt

**Keywords:** abiotic stress, gDNA cloning, *glyoxalase**I*, methylglyoxal, *Simmondsia chinensis*

## Abstract

Plant response to salt stress and the mechanism of salt tolerance have received major focus by plant biology researchers. Biotic stresses cause extensive losses in agricultural production globally, but abiotic stress causes significant increase in the methylglyoxal (MG) level of *Glyoxalase*
*I* (*Gly I*). Identification of salt-tolerant genes when characterizing their phenotypes will help to identify novel genes using polymerase chain reaction (PCR) to amplify the DNA coding region for *glyoxalase I*. This method is specific, requiring only genomic DNA and two pairs of PCR primers, and involving two successive PCR reactions. This method was used rapidly and easily identified *glyoxalase I* sequences as salt-tolerant genes from Jojoba (*Simmondsia chinensis* (Link) Schneider). In the present study, the *glyoxalase I* gene was isolated, amplified by PCR using gene-specific primers and sequenced from the jojoba plant, then compared with other *glyoxalase I* sequences in other plants and *glyoxalase I* genes like in *Brassica napus*, ID: KT720495.1; *Brassica juncea* ID: Y13239.1, *Arachis hypogaea*; ID: DQ989209.2; and *Arabidopsis thaliana* L, ID: AAL84986. The structural gene of *glyoxalase I*, when sequenced and analyzed, revealed that the uninterrupted open reading frame (ORF) of jojoba *Gly I* (*Jojo-Gly I*) spans 775 bp, corresponding to 185 amino acid residues, and shares 45.2% amino acid sequence identity to jojoba (Jojo-Gly I). The cloned ORF, in a multicopy constitutive expression plasmid, complemented the *Jojo-Gly I*, confirming that the encoded *Jojo-Gly I* in jojoba showed some homology with other known *glyoxalase I* sequences of plants.

## 1. Introduction

Jojoba (*Simmondsia chinensis* (Link) Schneider) is characterized as a desert economic, medical, wild, evergreen and perennial 100–200 year shrub [1]. In addition, the dicotyledonous and dioecious plant can bear adverse conditions, such as variation in temperature from −5 °C to 54 °C [2]. Jojoba is able to fight many diseases and pests and is characterized by its low water need, like date palms [3]. Therefore, it is ideal to be cultured on most of the Arabic region’s desert to take advantage of oil production that can be distinguished as a permeant, cheap fuel source to solve optimally the problem of energy in some Arabian countries with a lack of carbonic fuel [3]. Jojoba is used in the medical field as it has an important active constituent, Simmondssine, which is why jojoba’s scientific name is *Simmondsia chinensis* (Link) Schneider [4].

Some major tolerance mechanisms, including activities of late embryogenesis abundant proteins, are factors involved in signaling cascade damage induced by harsh temperatures [5,6]. The importance of a system in various cellular functions is associated with the metabolism of its physiological substrate and methylglyoxal. The formation of methylglyoxal in glycolytic biological systems is unavoidable. Moreover, under stress conditions, upregulation of enzymes involved in glycolysis and the TCA cycle (Tricarboxylic Acid Cycle) has been shown [5,7] and could lead to an increased production of methylglyoxal. In order to keep methylglyoxal concentration below a toxic level, an increase synthesis of glyoxalases is required. The role of glyoxalase enzymes was extensively investigated in various plant species which indicated their crucial role in salinity and drought, as well as heavy metal stress tolerance [8]. Therefore, upregulation of *Gly I* during the early period of plant development improves stress conditions and protection of plants against environmental stresses by overexpression of *Gly I*, resulting in the lowering of methylglyoxal levels. A further importance of the glyoxalase system is related to its cofactor i.e., glutathione (GSH). In physiological conditions, methylglyoxal rapidly reacts with GSH to form hemi-thioacetal. Decreasing the protein thiol levels in Methylglyoxal is shown by [9], as well as the levels of GSH in different types of cell [10,11]. The reduction of GSH levels in cells would have high repercussions, as it is essential [12]. Moreover, tolerance to xenobiotics and to biotic and abiotic environmental stresses suggests that enhanced GSH synthesis appears to be an intrinsic response of plants to stresses [13]. Decrease in the level of GSH could also alter redox sensitive regulatory proteins, such as thioredoxin and glutaredoxin, that could control the activities of enzymes with the job of cell differentiation and division of stresses [14]. Moreover, GSH is known to induce a variety of defense responses in plants [13]. Importance of glutathione in the cellular function was shown by manipulating the genes involved in regulation of glutathione homeostasis in transgenic plants, which resulted in increasing tolerance to oxidative stress [15,16,17,18].

The maintenance of a normal level of GSH under adverse conditions in the cells is an important physiological process. A number of enzymes including glutathione synthetase, glutathione S-transferase, glutathione reductase, glutathione peroxidase and dehydro-ascrobate reductase are known to be involved. Incidentally, it may be mentioned that identifing a cDNA clone which shows significant sequence homology with the type III class of GSTs is known to be stress inducible in nature. In an early attempt, the screening of the *Arabidopsis* expression library with glyoxalase I antibodies raised against *B. juncea* resulted in isolation of the *Gst* gene. The isolation of the *Gst* clone by immunological screening of the library with glyoxalase I antibodies in their study indicated a presence of common epitopes in both glyoxalase I and glutathione S-transferase proteins [8,19,20]. When the available sequences of Gly I from different organisms were used to generate a dendrogram, a segregation of glyoxalase I sequences into prokaryotic and eukaryotic types was obvious. Gly I sequence from Brassica, tomato, human and *Saccharomyces cerevisiae* seemed more closely related to each other. The glyoxalase I from *Salmonella* and *Escherichia coli* formed a separate group. Surprisingly, *Pseudomonas putida* glyoxalase I was dissimilar from the prokaryotic group.

The tomato *glyoxalase I* gene in the yeast expression system using the pYEX4 vector led to a 20-fold increase in the specific activity of the glyoxalase I enzyme. However, the sensitivity to methylglyoxal was similar to that of control cells [20].

Gayathri [21] explained the interrelationship between mangroves and their associated plants at the molecular level, and the adaptive mechanism of mangroves and associated plants under a saline condition. The identification of partial genomic DNA sequence coding for the *Glyoxalase I* gene in the white mangrove and its associated plant Ipomea, Macrantha, Roem and Schult indicated its role in the protection against methylglyoxal (MG) formed under various stresses and its future service for the development of various stress-tolerant crop plants.

The induced protein also showed cross-reactivity with antibodies raised against purified glyoxalase I protein from *B. juncea.* Moreover, the *E. coli* cells overexpressing *Gly I* conferred tolerance against high concentrations of methylglyoxal (20 mM). High specific activity of *B. juncea glyoxalase I* expressed in *E. coli* along with significant tolerance against methylglyoxal suggested that isolated clones indeed represented a *glyoxalase I* gene from *B. juncea glyoxalase I* genes isolated from other organisms that were also overexpressed in both prokaryotic and eukaryotic expression systems. The expression of the tomato *glyoxalase I* gene in the yeast expression system using a pYEX4 vector led to a 20-fold increase in specific activity of the glyoxalase I enzyme, however, the sensitivity to methylglyoxal-like control cells decreased [22].

It was surprising that even a 20-fold increase in specific activity of glyoxalase I enzyme in yeast was not able to protect the cells against methylglyoxal. In contrast to this, and similar to our results, overexpression of human *glyoxalase I* gene in *E.coli* or in the Cos cells resulted in a 110–180-fold increase in the glyoxalase I activity over the control, and these cells were tolerant to 5 mM of methylglyoxal [23,24]. Similarly, Rhee [25] showed that overexpression of *glyoxalase I* in *E. coli* cells lead to an increase in the specific activity (~150 fold) and this increase was able to impart tolerance against methylglyoxal. Maximum overexpression of *Gly I* gene (4000-fold increase in specific activity) was achieved for *P. putida glyoxalase I* in *E. coli* using the pBITac I vector [26]. Although different workers have achieved different levels of overexpression of *glyoxalase I*, all the overexpressing cells, except for yeast cells expressing the tomato *glyoxalase I* gene, were able to tolerate methylglyoxal. Therefore, the aim of this study was the sequencing and characterization of a *glyoxalase I* (*Jojo-Gly I*) gene from Jojoba in comparison with other *glyoxalase I* genes to use it as a salt-tolerant phenomena.

## 2. Materials and Methods

This experiment was carried out at Laboratories of Plant Biotechnology Department, National Research Centre in Egypt and Agricultural Biotechnology Department, Faculty of Agricultural and Food Sciences, King Faisal University, Al-Ahssa, Saudi Arabia.

Preparation of plant material as single node cutting (2 cm) explants were excised from a 5-year-old shrub from the Egyptian Natural Oil Company. Then explants were sterile to initiate cultures. Then the sterilization and primary cultivation of the excised explants (nodal segments and buds) were carried out as indicated in our previous research [27,28,29].

The plantlets under study were obtained through the callus resulting from the culture of the jojoba explants on the callus medium and then transferred to the shoot proliferation medium [30]

Initiation of aseptic cultures and culture establishment was carried out following the procedure of Solliman [28]. All explants were transferred on callus medium, then on-shoot proliferation medium was used, as followed by [30]. The basic Murashige–Skoog (MS) [31] medium concentration 4.4 g/L was supplemented by mg/L: 0.1 Myo-Inisitol, 160 AdSO_4_, 3 BAP, 1.5 Kin and 7000 Agar.

After adjusting the pH 5.7 ± 0.1, 40 mL of medium was dispensed into each 250 × 25 mm borosil rimless glass tube. The culture jars were wrapped and autoclaved at 1.05 kg cm^2^ and in 121 °C for 20 min., then the pH of the media at 5.7 was adjusted by the addition of 1 M NaOH. Cultures were incubated at 27 ± 2 °C under light provided by white fluorescent tubes, giving the intensity of about 2000 lux for 16 h/day. The explants were sub-cultured with the same media every three weeks.

### 2.1. Salt Concentration

This study was conducted in order to determine the effect of NaCl (200 mM NaCl to 1000 mM NaCl) for induction of salt stress. The control was treated with distilled water: (a) 0 mM NaCl (b) 200 mM NaCl; (c) 400 mM NaCl; (d) 600 mM NaCl; (e) 800 mM NaCl and (f) 1000 mM NaCl.

The solution of salt stress was prepared in levels in the water of tissue culture media. The suitable amounts of NaCl were mixed with distilled water and then detected by a portable Electrical Conductivity (EC) meter instrument. All jars were maintained under aseptic conditions. Foliar application was performed with the 1000 mM NaCl after 2 months of being used as the surfactant.

After that, seeds were germinated in culture media Murashige–Skoog (MS) which was supplemented with different hormones and NaCl.

### 2.2. Culture Condition

In all studied experiments, the cultured jars were incubated at 27 ± 1 °C. All the cultures treated were transferred to a growth room for 35 days. All the cultures were transferred to a growth room for 25 days, after that the culture was transferred and other physical and chemical properties were calculated.

### 2.3. DNA Cloning and Manipulation

DNA manipulations and purifications were performed according to [29] protocols. Putative jojoba *glyoxalase I* gene (*Jojo-Gly I*) from jojoba were cloned into the expression vector using Xba I and BamH I restriction endonuclease enzymes. Construction of *Jojo-Gly I* gene was done in pSC-A (StrataClone PCR Cloning kits, 11011 North Torrey Pines Road, La Jolla, CA, USA). The *Jojo-Gly I* gene was PCR-amplified by using the Glo1f primer–5′ (5′-CCGAGATCTAAAGCTTTTTCAATGGGGATC-3′) and the Glo1r primer–3′ (5′-TGTGAGATCTAAGCTTGAACGCAT-3′) primers. The product (~775 bp) was cloned into the pSC-A (StrataClone PCR Cloning kit, Stratagene) vector and sent for sequencing.

### 2.4. DNA Cloning and DNA Sequencing

DNA cloning and plasmid constructions were done as described by Sambrook [32]. DNA was sequenced by the chain termination method at Macrogen, 0F, 254, Beotkkot-ro, Geumcheon-gu, Seoul, Korea. For the sequencing of the *Jojo-Gly I* gene, vector primers, cloning primers, and different sequencing primers were used.

### 2.5. Ligation of the PCR Product with the pSC-A Vector

Many copies of the *Jojo-Gly I* gene, the glyoxalase I salt tolerance gene from Jojoba, were made before they were inserted into plasmids using PCR followed by the ligation of these PCR products with a pSC-A vector. This vector has 3′T overhangs that will bond with the 5′A overhangs that get added to the ends by thermostable polymerases. This ligation product was transformed into competent *E. coli* DH5α, then plated on selective media. Positive clones were confirmed by restriction analysis and sequencing. Jojoba *glyoxalase I* gene (*Jojo-Gly I*) sequence was analyzed.

#### Preparation of Competent Cells and Transformation

Competent *E. coli* DH5α cells were made according to protocols of Hanahan [33]. Verification by restriction digestion was performed on the abovementioned eluted *E. coli* to check for the presence of the correct plasmid.

### 2.6. Purification of DNA Fragment from Agarose Gel

After overnight digestion at 37 °C, the digested product was electrophoresed on an Agarose gel, and 1 μL loading buffer was added to 5 μL digest and loaded on a 0.7% (w/v) agarose gel in the Tris-acetate-EDTA (TAE) buffer. The gel was run for 180 min at 85 V. By using a standard molecular wt. marker (1 Kb ladder), the desired fragment was identified, cut, and purified by the following methods.

### 2.7. Qiaquick Gel Extraction Protocol (from Qiagen)

To the cut pieces of agarose gel containing DNA fragments, 3 volumes of buffer (as supplied with Qiaquik gel extraction kits, QIAGEN, 40474 Düsseldorf, Germany) were added and dissolved by heating to 50 °C for 10 min. The mixture was loaded onto Qiaquick spin column and spun briefly. The flow-through was discarded and the column was washed twice with a PE buffer. The purified DNA fragment was eluted with 50 µL (10 mM) Tris-HCl pH 8.0, then purified fragments were ligated with digested plasmids using T4 ligase O/N.

### 2.8. Homology and Structural Comparison of Jojo-Gly I Gene from Jojoba

Most of the sequence (protein and DNA) analyses were performed using CLC vector program (Software company headquartered in Aarhus, Denmark, and with offices in Cambridge, Massachusetts, Tokyo, Taipei and Delhi) and Genbank database. Homology searches were done using FASTA (NIH, USA) and multiple sequence alignment was done using CLUSTALW (NIH, USA).

Isolation of plasmid DNA: This method was adopted from Sambrook et al. [32].

### 2.9. DNA Sequence Analysis

Genomic and cDNA clones were subcloned into pSC-A (StrataClone PCR Cloning kit, Stratagene). Clones for sequencing were generated by either exonuclease III-Henikoff [33] or restriction-enzyme-generated deletions. Single- and double-stranded DNA sequencing was conducted by the dideoxynucleotide chain termination method [34] using the Sequenase system. All regions were sequenced on both strands. Analysis of DNA sequences was conducted using DNASTAR (Aarhus, Denmark).

Intron/exon boundaries were identified by comparing genomic and cDNA sequences. To identify *glyoxalase I* sequences, BLAST searches were conducted on the GenBank. Glyoxalase I protein sequences were aligned using the Clustal method, which is contained in the DNASTAR package.

Sequence alignments and percent identities were computed using CLC Main Workbench (version 8.1.2) (http://www.qiagenbioinformatics.com).

The data was statistically analyzed by analysis of variance (ANOVA) for the completely randomized design. The treatment means were compared using Least Significant Difference (LSD) at a 5% level of probability [35]. All computations and statistical analysis were performed using the computer and SAS software package (SAS, based in Cary, NC, USA) (2005). Analysis of variance related to CR experiments was conducted as described by [36].

## 3. Results and Discussion

The present work was undertaken to study the role of *glyoxalase I* gene (*Gly I*) in plants. Our strategies included isolation of *glyoxalase I* gene from jojoba plant (*Jojo-Gly I*), regulation of *glyoxalase I* during development and environmental stress conditions, and upregulation or downregulation of *glyoxalase I* gene expression in jojoba plants following isolated *glyoxalase I* clone in the sense of direction, respectively.

### 3.1. Effect of Abiotic Stress Induced by NaCl on Growth of Jojoba Plant Tissue Culture

Jojoba cultures were determined and related to response as a result of adding to different concentrations of NaCl which were supplemented to media as follows: 0, 200, 400, 600, 800 and 1000 mM on some physical properties such as fresh weight, dry weight, leaf number, plant length, and shoot numbers. Comparison of our studies with many reports in plant tissue culture in vitro have been done to explain or evaluate the salt tolerance on jojoba plants in response to different concentrations of NaCl. Under the tissue culture conditions, few studies of the factors affecting jojoba, such as media structure, light period, and other salt stress, have been conducted [37]. As well, growth regulators have always had an effect on the morphogenic jojoba cultures.

Some of the morphogenesis characteristic response seems to be dependent on plant growth regulators, which were added to the nutrient medium; another reason is the plant cultivar. In this study, the abovementioned data from Figure 1 indicate that the effects of adding different levels from NaCl to MS medium varied from one concentration to another. This is clear that growth of jojoba has the lowest value of NaCl (control) and had no effect on jojoba growth in MS mediums with different levels of NaCl, but increasing the levels of NaCl from 200 mM up to 1000 mM was effective on jojoba growth. It could be concluded from the results that highest value of NaCl is very effective on jojoba growth. The higher values of fresh weight were obtained with low level of concentration of NaCl compared with other treatments. Control treatments were recorded (as shown in Table 1), although the lowest value was recorded with the 1000 mM of NaCl.

Salt significantly influenced plant height, branch and leaf numbers per plant, and color of jojoba plant. The stress done via salinity with Na salts not only reduces Ca availability, it also reduces the move of Ca and its mobility for growing parts in the plant. This action reverts to the plant in the quality and vegetative organs [20,38]. Spinach tomato affected with high concentrations of NaCl induced K defiance [39,40]. Moreover, high salinity induced N insufficiency in tomato, cabbage, and lettuce. Averaged data showed that the shortest plants were those subjected to salt stress treatments (Figure 1).

From the above results, we started looking for the genes involved in salt tolerance in the jojoba plant. The next part was focused on determining which genes are involved in the molecular mechanisms of tolerance in jojoba.

### 3.2. PCR Amplification of Jojoba Glyoxalase I (Jojo-Gly I) Gene

In this study, we can understand the physiological significance of the glyoxalase system in certain plants, as the gene for *glyoxalase I* was isolated and characterized from the Jojoba plant. PCR amplification of targeted PCR *glyoxalase I* gene fragments from jojoba genomic DNA with *glyoxalase I* (*Jojo-Gly I*) of the gene confer tolerance in plants under stress from jojoba (Figure 2 and Figure 3); a photo showing some difference in the DNA size also shows the sequencing in Figure 4, Figure 5, Figure 6 and Figure 7. A *glyoxalase I* gene DNA from jojoba was constructed in a pSC-A vector from DNA isolated from leaves of jojoba which were stressed with NaCl (Figure 8a,b). Using a highly specific *glyoxalase I* primer, which was created from the Genbank data for *glyoxalase I* of *Brassica juncea*, we were able to isolate a full length of *glyoxalase I* clone, which showed very high homology with the other reported *glyoxalase I* gene from the other plants and mammalian, as well as microbial systems. This genomic *glyoxalase I* gene was found to code for a protein of 195 amino acids with a calculated as mol per mass of 22.782 and pI of 5.87. The fragment was cloned into the pSC-A easy vector with site-specific recombination. Recombinant plasmid pSC-A was confirmed by digestion of Xba I and BamH I Figure 9 and by sequencing.

### 3.3. Cloning the 5’ End of the Jojo-Gly I

The 5’ end of the *Jo-Go1* gene was isolated using the gene-specific primer gly-f-jojoba (5’-AGGATCAACGACTCCAAC-3’ as the forward primer with the gene-specific primer gly-r-jojoba 5’-GAGAGATATTGCGCGTAGATAG-3’ as a reverse primer. PCR amplification used jojo-gly1 primers and the primer resulted in a 775 bp fragment that was cloned into pSC-A (StrataClone PCR Cloning kit, Stratagene) and verified by sequence analysis.

The 5’ end of *glyoxalase I* from jojoba plants was analyzed by conducting 5’ RACE, and then the amplification products were cloned into pSC-A (StrataClone PCR Cloning kit, Stratagene) and verified by sequence analysis. Besides, screening of *glyoxalase I* of jojoba and *Brassica* also resulted in identification of partial genomic and *glyoxalase I* clone which showed significant homology with the type III class of GSTs. This class of GSTs are known to be regulated by various stresses. Both the genes have been submitted in Genbank with the accession no. Y13239 (for cDNA coding for *glyoxalase I*) and Y13829 (for cDNA coding for GST). We also attempted to isolate *glyoxalase I* gene from tobacco. However, screening of Jojoba *glyoxalase I* led to the isolation of a DNA clone which showed 95% homology with a gene cloned from Arabidopsis, which has been suggested to code for GST.

### 3.4. BLAST Search for Identical Protein Sequences

It was concluded that protein levels showed more homology with Gly1 from the *Brassica juncea* region at the nucleotide level due to degeneracy of the genetic code. Thus, pairwise comparison at the amino acid level is more reliable than nucleotide–nucleotide comparison for establishing the identity of any *Gly1 gene*.

The nucleotide sequences were translated into polypeptides using the ExPASy translate tool and an identity search was made using the BLASTP algorithm. Two *Jojo-Gly I* gene analogs showed high similarity with *Brassica juncea* deposited in the GenBank (Accession number: embCAA73691.1, Figure 6 and Figure 7).

The ubiquitous glyoxalase system was involved in the detoxification of methylglyoxal, a cytotoxic compound produced non-enzymatically as well as enzymatically from the triose phosphates as a byproduct of Embden-Meyerhof and polyol pathways, and during metabolism of amino acids and acetone [41,42]. Studies on the role of the *glyoxalase* gene in animal and microbial systems have been implicated to reveal cellular functions including cell division/proliferation, microtubular assembly, vesicle mobilization, tumor growth, and clinical complications associated with diabetes mellitus and various diseases [41]. Studies on the plant glyoxalases have also related the presence and modulation of glyoxalase activities to cell division/proliferation, and its regulation in response to certain external factors including light, hormone, and environmental stresses [22,43,44,45]. Most of these studies are rather preliminary and hence a definite role of this system in plants is not yet assigned to this system.

To understand the physiological role of glyoxalase system in plants, we have cloned and characterized a *glyoxalase I* gene from *B. juncea*. We found that the expression of *glyoxalase I* gene increased significantly in response to abiotic environmental stresses such as salt, heavy metal, and overexpression of *Gly I* gene led to the protection of plants against salt and heavy metal stresses. The present study favors an essential role of a ubiquitously expressed protein like glyoxalase I under stress conditions.

### 3.5. Phylogenetic and Glyoxalase I Analyses Show the Origin of Glyoxalase I Plant Genes Derived from Jojoba and Other Plants

Phylogenetics of *glyoxalase I* gene from plants: An analysis of nucleotide sequences from the *glyoxalase I* gene from jojoba with the other sources. Phylogenetic analyses of complete *glyoxalase I* gene sequences provide much-improved confidence in the relationships among major lineages of the jojoba *glyoxalase I* gene to plant *glyoxalase I* gene and provide a framework for other *glyoxalase I* genes like in *Brassica napus*, ID: KT720495.1; *Brassica juncea* ID: Y13239.1, *Arachis hypogaea* ID: DQ989209.2; and *A. thaliana* L, ID: AAL84986.

### 3.6. Multiple-Sequence Alignment and Phylogenetic Analysis

Phylogenetic analysis of *Jojo-Gly I* from jojoba and *Gly I* genes were performed separately using all reported members from *Arabidopsis*, *Brassica napus,* and *Arachis hypogaea* as shown in Figure 10. The tree was constructed using full-length amino acid sequences based on a neighbor-joining method with 100 bootstrap value.

The presence of the multi-stress responsive *Gly I* gene indicates its role in the protection against MG; this study also reveals the interrelationship between jojoba plants at the molecular level and adaptive mechanisms of jojoba plants under saline conditions. Previously, genome-wide analysis of *glyoxalase* genes has been conducted in model plants *Arabidopsis*, rice and legume species, but no such study was performed in the jojoba plant. In addition to sequence comparison, we confirmed that the *B. juncea* cDNA that we have isolated is coding for glyoxalase I. Functional analysis of the isolated *Gly I* clone was done by overexpressing the coding region in the isopropyl-D-thiogalactopyranoside (IPTG)—inducible E. coli expression system using a pRSETA-Gly I vector. The expression of protein showed high glyoxalase I activity (1200-fold increase when compared to control).

In conclusion, axillary shoot proliferation from the nodal explants of jojoba varied considerably at different growth regulator concentrations in the medium, and MS basal medium with the levels of NaCl (NaCl from 800) showed bud-break in 20% (1.0 nodes/shoot) of the cultures compared to 5% (1.0 nodes/shoot) cultures at levels of NaCl (NaCl from 600). Most of the nodal explants turned brown at MS concentration with all levels of NaCl after five weeks of incubation; hence, for establishing nodal segment cultures, the MS medium was used initially. Glyoxalase I was a part of the glyoxalase system present in the cytosol of cells. Moreover, under stress conditions, the upregulation of enzymes was involved in glycolysis and led to increased production of methylglyoxal. To keep methylglyoxal concentration below the toxic level, an increased synthesis of glyoxalases is required. Therefore, *Gly I*’s upregulation during the early period of plant development and during stress conditions and protection of plants against environmental stresses by overexpression of *Gly I* results in the lowering of methylglyoxal levels. Further importance of the glyoxalase system is related to its cofactor, i.e., GSH. In physiological conditions, methylglyoxal rapidly reacts with GSH to form hemithioacetal. Methylglyoxal has been shown to decrease the protein thiol levels [46]. The methylglyoxal is a major physiological substrate for glyoxalase I, and its accumulation increased markedly when glyoxalase I was inhibited in situ by cell-permeable glyoxalase I inhibitors and by depletion of GSH [41,47,48,49]. On the other hand, Glyoxalase I activity prevents the accumulation of these reactive α-oxoaldehydes and thereby suppresses α-oxoaldehyde-mediated glycation reactions. The mechanism of GSH production is a central role in various anti-oxidative processes and xenobiotic detoxification [50].

These results might have a potential in agriculture productivity and expanding agricultural land to secure food globally. This increase was able to impart tolerance against methylglyoxal. Maximum overexpression of the *Gly I* gene (4000-fold increase in a specific activity) was achieved for *P. putida glyoxalase I* in *E. coli* using a pBITac I vector. Although different workers have achieved different levels of overexpression of *glyoxalase I*, all the overexpressing cells, except the yeast cells expressing tomato *Gly I*, tolerate methylglyoxal. Moreover, the tolerance level to methylglyoxal depended on the level of *glyoxalase I* [20]. Further experiments will be needed to understand the relationship between gene identification with plant metabolism and physiology by determining some metabolites.

## Figures and Tables

**Figure 1 plants-09-01285-f001:**
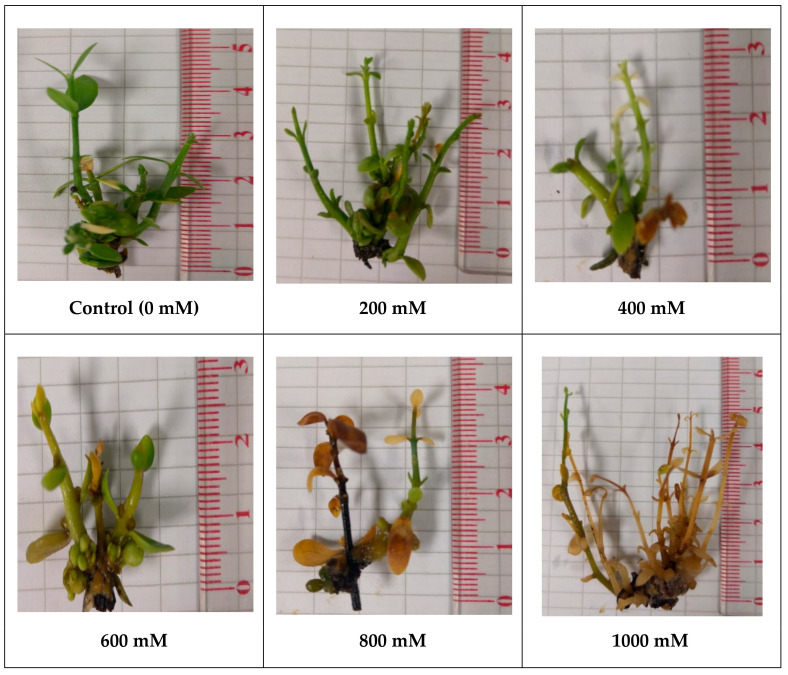
Treated jojoba plants growing on Murashige–Skoog (MS) medium with different levels of NaCl (0, 200, 400, 600, 800 and 1000 mM of NaCl) completed with double distal water.

**Figure 2 plants-09-01285-f002:**
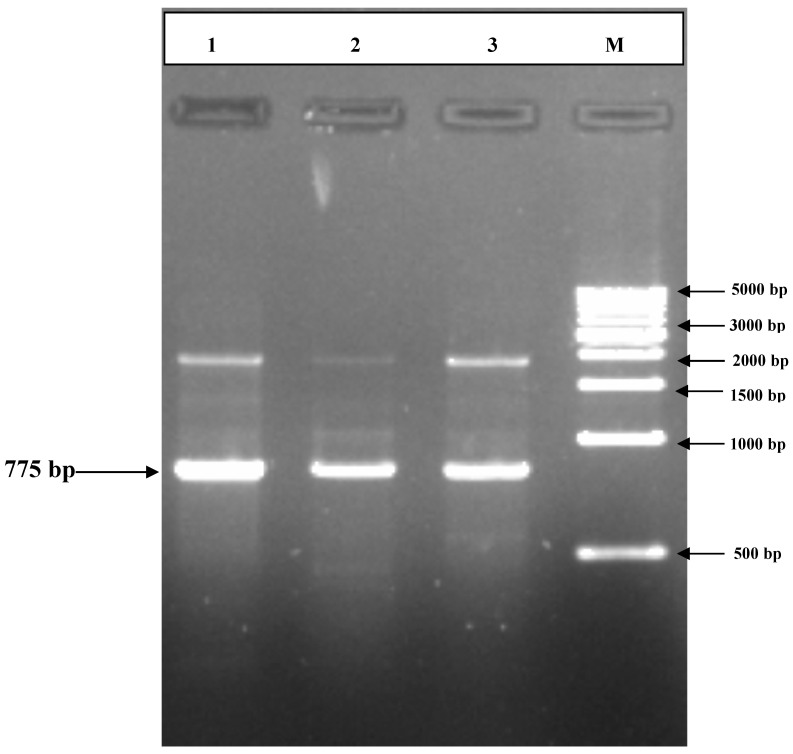
Agarose gel electrophoresis showing the PCR products of *glyoxalase I* gene region of jojoba using the PCR system as shown in lanes (1, 2 and 3). PCR products of correct size (775 bp) were respectively amplified using Jojo-gly1f and Jojo-gly1R primers.

**Figure 3 plants-09-01285-f003:**
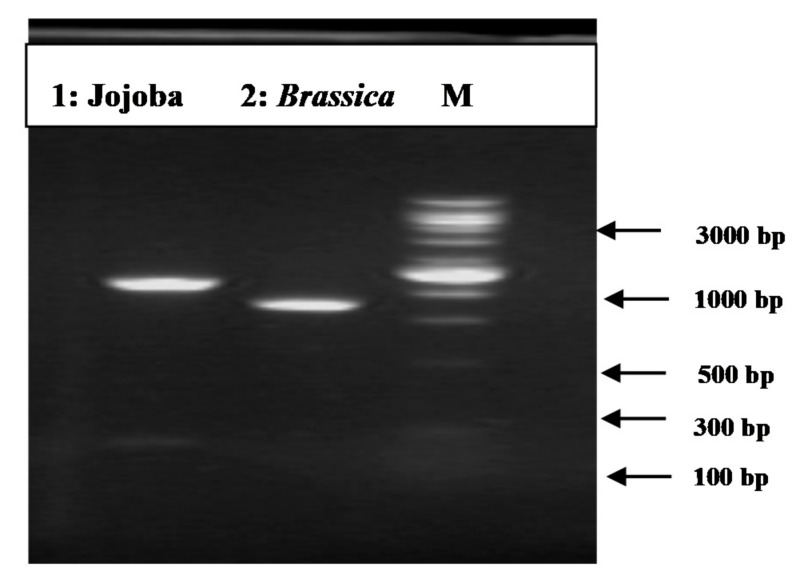
PCR amplification of targeted PCR fragments in Lane (1) from jojoba (*Jojo-Gly I*) with *glyoxalase I* (*Gly I*) of the gene confer tolerance in plants under stress compared with Lane (2) from *Brassica juncea* (Accession number: embY13239.1). The “100 bp ladder” was used as a size marker.

**Figure 4 plants-09-01285-f004:**
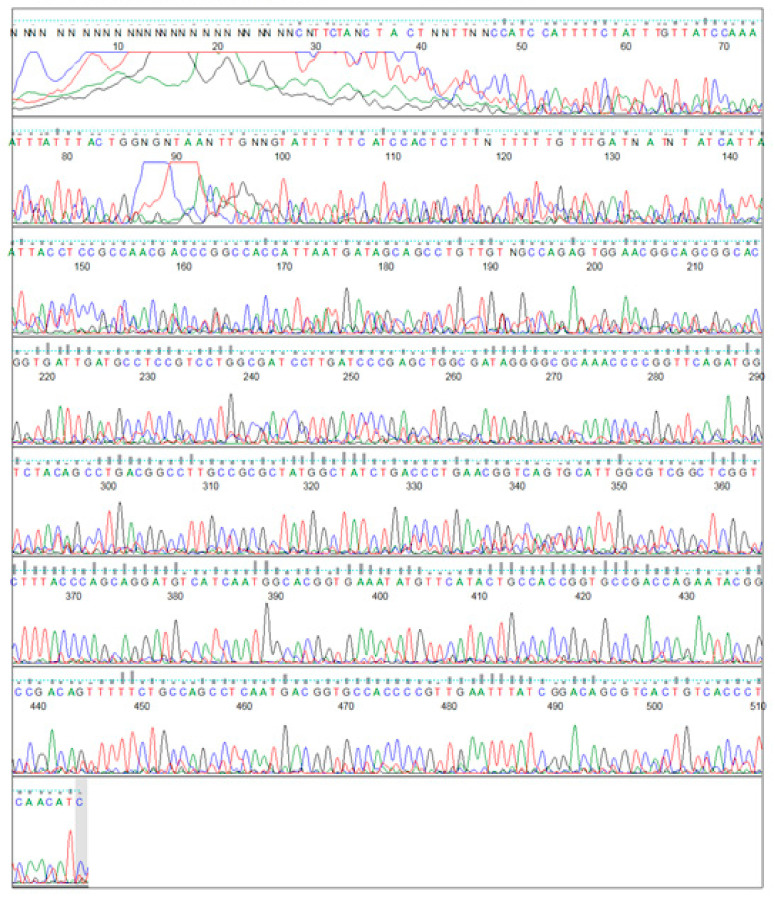
Gene sequencing of the *glyoxalase I*, a salt tolerance gene from Jojoba (*Jo-Glo1*), by forward primer sequence.

**Figure 5 plants-09-01285-f005:**
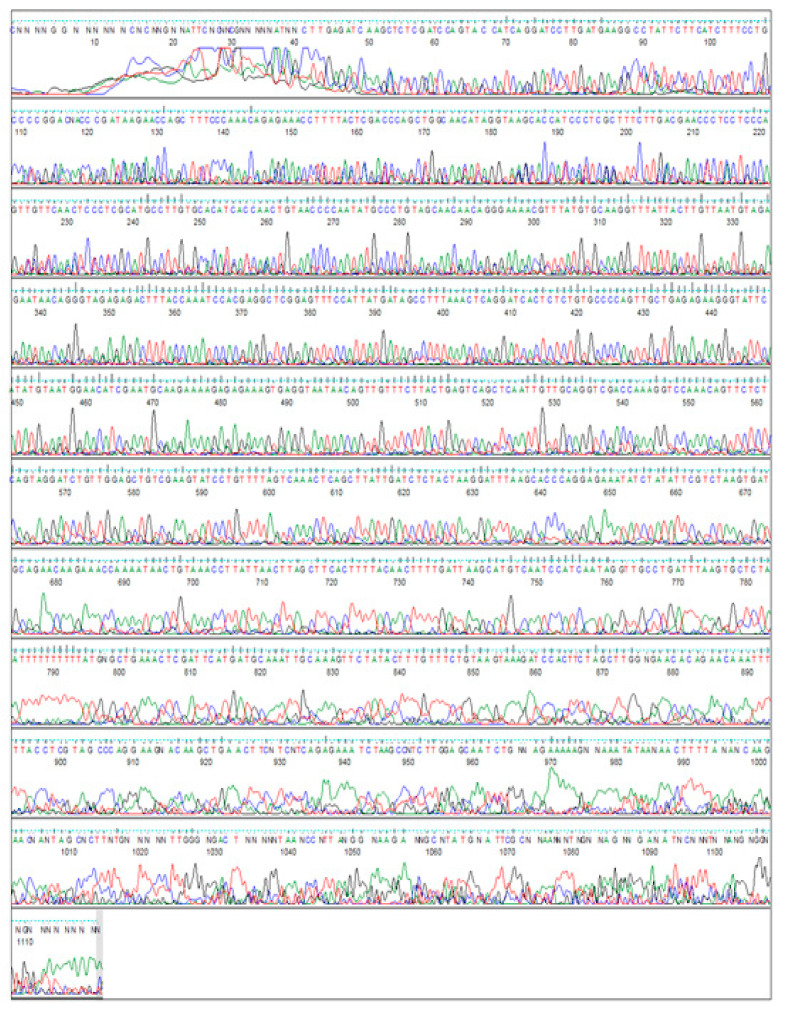
Gene sequencing of the *glyoxalase I*, a salt tolerance gene from Jojoba (*Jo-Glo1*), by reverse primer sequence.

**Figure 6 plants-09-01285-f006:**
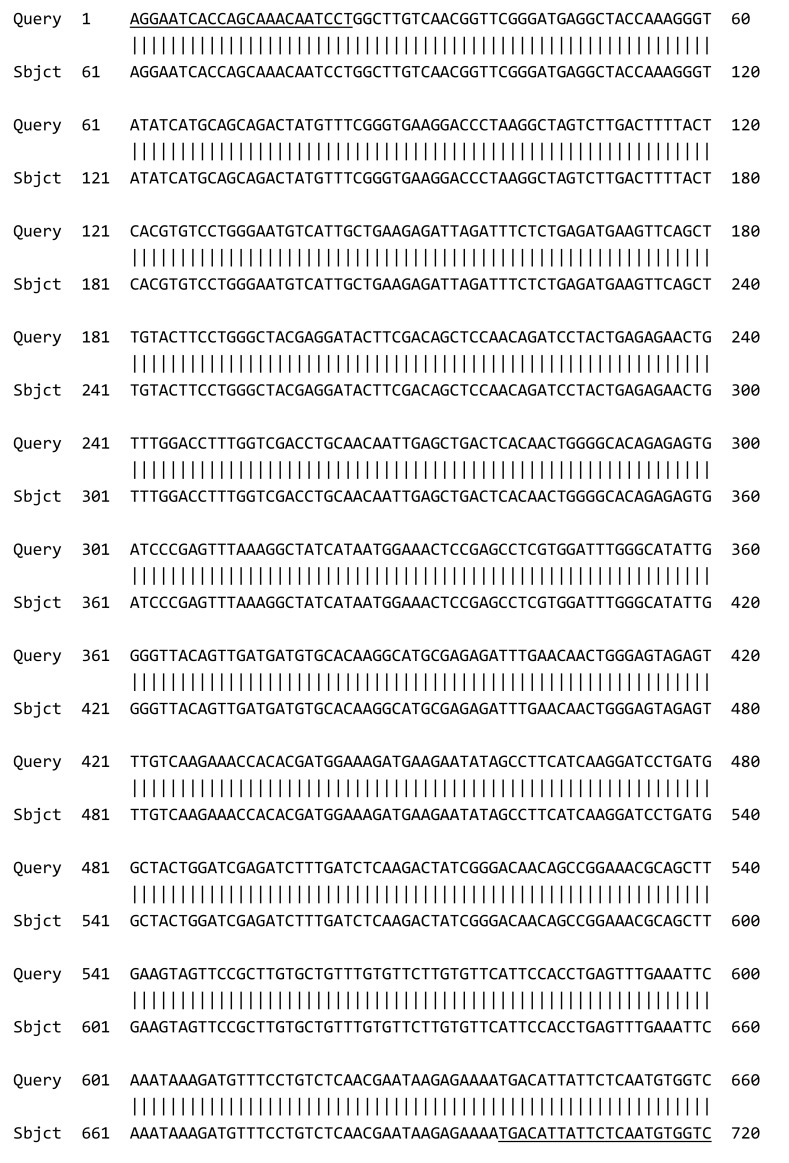
Nucleotide sequence of *Jo-glo1* from jojoba with *gly-1* (*glyoxalase I*) of the coding region of the gene confers tolerance in plants under stress; *glyoxalase I* gene from *Brassica juncea* (Accession number: embY13239.1, *glyoxalase I* Length = 775). The underlined regions correspond to the primers used for PCR amplification.

**Figure 7 plants-09-01285-f007:**
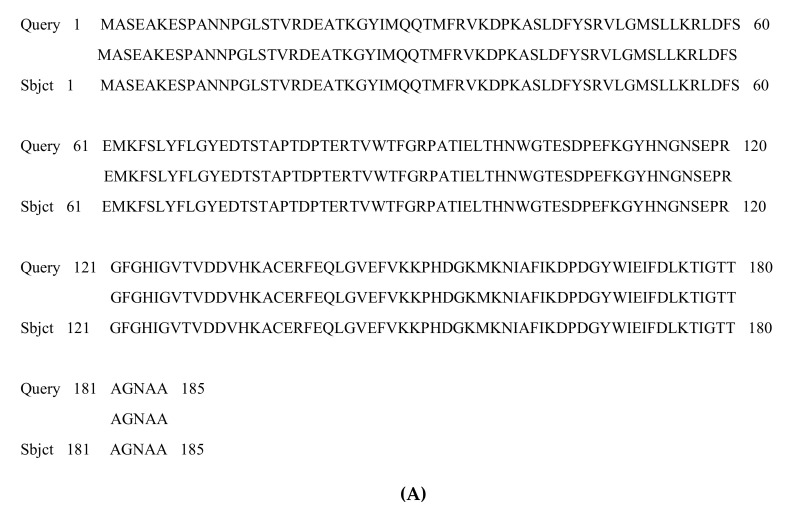
Sequence analysis of jojoba *glyoxalase I* gene. (**A**) Comparison of the predicted amino acid sequences of *glyoxalase I* from jojoba (*Jojo-Gly I*), with *glyoxalase I* gene conferring tolerance to plants under stress, *glyoxalase I* gene from *Brassica juncea* (Accession number: emb CAA73691.1). The deduced amino acid sequence of the *glyoxalase I* gene product is shown and the amino acids that differ from the *glyoxalase I* gene product are indicated. (**B**) Multiple sequence alignment of jojoba *glyoxalase I* (*Jojo-Gly I*) was carried out using Blast. Comparison of the nucleotide sequences of *Jojo-Gly I* from jojoba with *glyoxalase I* sequence of *Arabidopsis thaliana* lactoylglutathione lyase family protein/glyoxalase I family protein (AT1G08110), Mrna. Sequence ID: NM_001035918.2 Length: 1021 Number of Matches: 2, calculates the statistical significance.

**Figure 8 plants-09-01285-f008:**
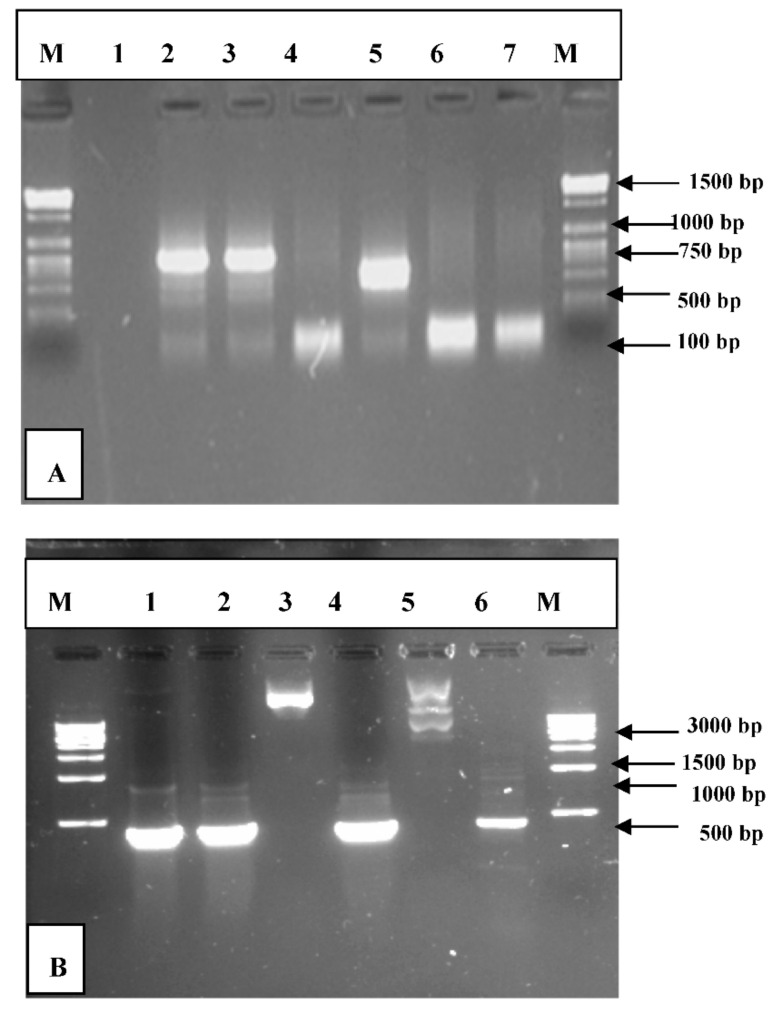
Agarose gel electrophoresis showing the PCR products after cloning into pSC-A vector using PCR system. PCR was performed on a recombinant *Jojo-Gly I* clone to verify the presence of *Jojo-Gly I* gene insert in a pSC-A vector. (**A**) Lanes (2, 3 and 5) show the expected size of the amplified PCR product using M13 universal primers, and (**B**) Lanes (1, 2, 4 and 6) show the expected size of the amplified PCR with specific primers by M13 and gly1r.

**Figure 9 plants-09-01285-f009:**
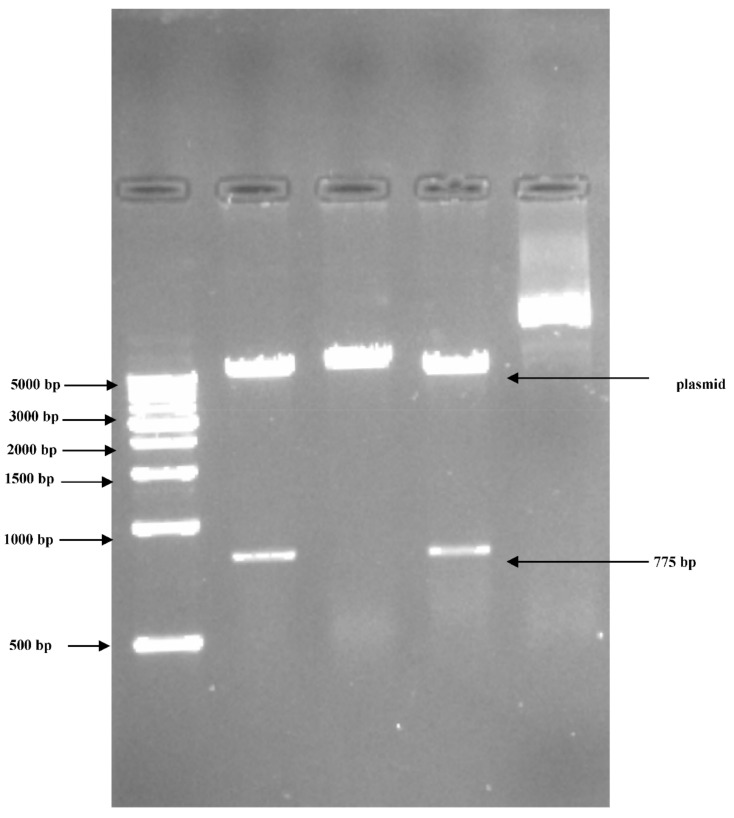
Restriction digestion products for the clone of *Jojo-Gly I* after cloning the *Jojo-Gly I* in pSC-A vector and digestion by Xba I and BamH I.

**Figure 10 plants-09-01285-f010:**
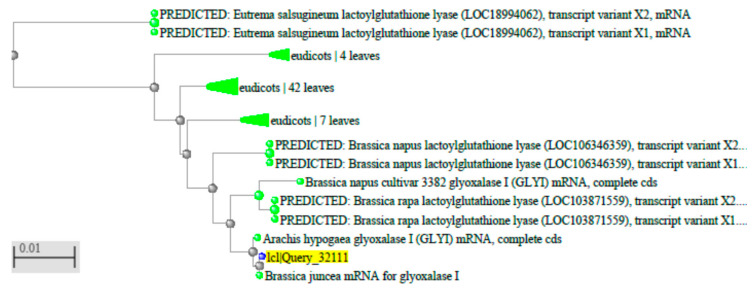
Phylogenetic relationships between the *Jojo-GlyI* gene using the gene structure predictions and conversed protein motifs. Phylogenetic analysis of *glyoxalase I* gene in jojoba, *Arabidopsis*, *Brassica napus*, *Arachis hypogaea*.

**Table 1 plants-09-01285-t001:** Treated jojoba plants growing on MS medium with different levels of NaCl (NaCl from 400, 600, 800 and 1000 mM of NaCl). Shoot number, leaf number, and plant length were determined in the table and in general.

Treatments NaCl (mM)	Number of Jars	Shoots per Culture	Number of Shoots	Length of Shoots (cm)	Stem Diameter (mL)	No. of Leaves
Control	3	5	39.33a	2.40ab	2.00b	58.33a
200	2	5	24.00b	2.50ab	2.00b	32.00b
400	3	5	23.67b	2.93a	2.00b	31.67b
600	2	4	10.33d	1.85c	1.50c	18.67c
800	2	4	19.00c	2.30ab	3.00a	29.00b
1000	3	3	17.33c	2.77a	3.00a	26.67b

a, b, c and d (evaluation of results in descending order).

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
