# Peer review of "Salt-Tolerant Phenomena, Sequencing and Characterization of a Glyoxalase I (Jojo-Gly I) Gene from Jojoba in Comparison with Other Glyoxalase I Genes"

_plants, 2020, doi:10.3390/plants9101285_

Round 1

Reviewer 1 Report

The authors improved the manuscript. 

But still I would like to reccomend to check some moments:

Lines 169-173, Lines 190-193. Сheck spaces between lines.

Line 220. Is it nessessary to use " before word fresh ....?

Lines 243-244. ... tomato, Cabbage and Lettuse should be change to tomato, cabbage and lettuse.

Table 1. would be better to give ± standart deviations next to presented numbers or indicate by letters significant differences bettwen variants.

In general, after making the above-mentioned comments, the paper may be accepted for publication.

Author Response

Dear Editor,

Thank you for your positive feedback and we have amended the manuscript according to the comments of Rev.1 as follows:

But still I would like to reccomend to check some moments:

Lines 169-173, Lines 190-193. Сheck spaces between lines.

Corrected 

Line 220. Is it nessessary to use " before word fresh ....?

Corrected

Lines 243-244. ... tomato, Cabbage and Lettuse should be change to tomato, cabbage and lettuse.

Corrected

Table 1. would be better to give ± standart deviations next to presented numbers or indicate by letters significant differences bettwen variants.

Amended

In general, after making the above-mentioned comments, the paper may be accepted for publication.

Thank you

Reviewer 2 Report

The manuscript entitled "Salt tolerant phenomena, sequencing and characterization of a glyoxalase I (Jojo-Gly I) gene from Jojoba in comparison with other glyoxalase I genes" is basically the description of the cloning of the glyoxalase I gene and some characterization of the salt tolerance of Jojoba.

The manuscript is poorly written, hard to read and to understand. English is not up to standard. Just the cloning of a gene is not sufficient for publication in Plants. The manuscript does not present any hypothesis.

At the end of the introduction the authors state the aim of this work nd confirm that the only scope was to clone the glyoxalase I gene:

"Therefore, the aim of this study to sequencing and characterization of a
glyoxalase I (Jojo-Gly I) gene from Jojoba in comparison with other glyoxalase I genes and use it as a salt tolerant phenomena".

In addition the final sentence: "use it as a salt tolerant phenomena" does not mean anything

Author Response

Reviewer 2 dose not provided actual scientific comments dealing with manuscript contents.

Reviewer 3 Report

This version is well revised.

Author Response

Thank you

This manuscript is a resubmission of an earlier submission. The following is a list of the peer review reports and author responses from that submission.

Round 1

Reviewer 1 Report

The authors improved paper according to previous comments. However, the paper still requires to be revised for spellings and stylistic errors throughout the text. Also need to check again the fonts in the text (For example, the font size of Fig 9A description is different from other Figures.....). 

Reviewer 2 Report

The authors tried to improve the manuscript but they have not taken care of many vital suggestions. They could not make a relationship of the studied genes with plant metabolism and physiology. I suggested to measure some metabolites but they failed.

The discussion is also very poor.

They have overlooked many vital papers on MG detoxification and Glyoxalase system,

Overall, it is very poorly written.